# Structure, dynamics and transferability of the metal-dependent polyhistidine tetramerization motif TetrHis for single-chain Fv antibodies

Robert D. Healey[1,3], Louise Couillaud[1], François Hoh[2], Assia Mouhand [2], Aurelien Fouillen [1], Pierre Couvineau [1], Sébastien Granier [1✉] & Cédric Leyrat [1✉]

The polyhistidine (6XHis) motif is one of the most ubiquitous protein purification tags. The 6XHis motif enables the binding of tagged proteins to various metals, which can be advantageously used for purification with immobilized metal affinity chromatography. Despite its popularity, protein structures encompassing metal-bound 6XHis are rare. Here, we obtained a 2.5 Å resolution crystal structure of a single chain Fv antibody (scFv) bearing a C-terminal sortase motif, 6XHis and TwinStrep tags (LPETGHHHHHHWSHPQFEK[$G_3S$]$_3$WSHPQFEK). The structure, obtained in the presence of cobalt, reveals a unique tetramerization motif (TetrHis) stabilized by 8 $Co^{2+}$ ions. The TetrHis motif contains four 6 residues-long β-strands, and each metal center coordinates 3 to 5 residues, including all 6XHis histidines. By combining dynamic light scattering, small angle x-ray scattering and molecular dynamics simulations, We investigated the influence of $Co^{2+}$ on the conformational dynamics of scFv 2A2, observing an open/close equilibrium of the monomer and the formation of cobalt-stabilized tetramers. By using a similar scFv design, we demonstrate the transferability of the tetramerization property. This novel metal-dependent tetramerization motif might be used as a fiducial marker for cryoelectron microscopy of scFv complexes, or even provide a starting point for designing metal-loaded biomaterials.

[1] IGF, University of Montpellier, CNRS, INSERM, Montpellier, France. [2] Centre de Biologie Structurale, University of Montpellier, CNRS, INSERM, Montpellier, France. [3] Present address: Sosei-Heptares, Steinmetz Building, Granta Park, Cambridge CB21 6DG, UK. ✉email: sebastien.granier@igf.cnrs.fr; cedric.leyrat@igf.cnrs.fr

Polyhistidine tags are peptides consisting of 6 or more consecutive histidine residues (6XHis) that are commonly used for protein purification by immobilized metal affinity chromatography (IMAC) due to their metal binding properties[1]. These polyhistidine sequences bind various metal ions such as $Co^{2+}$, $Ni^{2+}$, $Zn^{2+}$, and $Cu^{2+}$ and are usually genetically fused to the N- or C-terminus of recombinant proteins. Histidine-rich motifs (including homopolymeric histidine tracts) also occur in diverse natural proteins and peptides[2,3], such as various zinc transporters[4], bacterial nickel metabolism proteins[5,6], human transcription factors[7] or snake venoms[8]. A number of isolated peptide sequences have been studied for their metal-binding properties, ranging from a simple 6XHis[9] to more complex sequences such as the poly-His/poly-Gly peptide pHpG-1 ($EDDH_9GVG_{10}$) isolated from the venom of the African viper *Atheris squamigera*[8,10,11], or the peptide $H_2ASHGH_2NSH_2PQH_{11}$ corresponding to residues 33–57 of human Forkhead box protein G1[7]. These peptides form complexes with metal ions that usually display polymorphic binding states[7–9,12], sometimes coupled with the stabilization of α-helical structure[8].

A sequence motif search of protein structures containing a 6XHis in the PDB yields ~42k results, indicating the sequence is highly prevalent in constructs used for structure determination. However, the 6XHis motif usually behaves as an intrinsically disordered region, adopting various conformations, and is thus extremely rare in electron density maps. For this reason, a structure motif search for 4 consecutive histidines yields only 250 structures. Restricting the search to structures containing any zinc, copper, nickel, cobalt, iron, or cadmium returns 81 entries, i.e. 0.002% of entries where the 6XHis was present.

In this study, we report the serendipitous discovery of TetrHis (acronym for Tetrameric His-Tag), a metal-dependent tetramerization motif with minimal sequence ETGHHHHHHWSHPQ observed in the cobalt-bound crystallographic structure of a single-chain variable fragment (scFv) antibody. ScFvs are composed of the variable regions of the antibody heavy and light chains ($V_H$ and $V_L$ domains) connected by a flexible peptide linker, and they represent the smallest engineered antibody fragment containing the parental specificity. ScFvs can display significant inter-domain flexibility between $V_H$ and $V_L$ domains[13] and have a tendency to form dimers and higher order oligomers depending on the length of the linker peptide[14–16]. However, the identification and characterization of a metal-dependent tetramerization motif utilizing a polyhistidine sequence in scFvs represents a novel and unexplored area of research. The sequence of the TetrHis motif corresponds to a 6XHis tag with a preceding ETG sequence from a sortase motif (LPETG) and a trailing sequence WSHPQ from a TwinStrep tag that were fused at the C-terminus of the expressed scFv construct (Fig. 1a). The scFv used in this work, 2A2, was designed by grafting the complementarity determining regions (CDRs) of the 2A2 antibody described to bind the lipid ceramide[17] onto a murine scFv scaffold with a [GGGGS]₄ linker (taken from PDB entry 5LX9 chain B[18]), with the initial aim of studying its binding to ceramide. Below we describe the crystal structure of scFv 2A2 in its cobalt-stabilized tetrameric state that revealed the TetrHis motif. We next characterize the dimensions, structure, and dynamics of the tagged scFv molecule in the absence or presence of $Co^{2+}$ ions using a combination of dynamic light scattering (DLS), small angle x-ray scattering (SAXS), molecular dynamics simulations (MDS), and ensemble analysis, revealing its conformational equilibria in atomistic details. Finally, we use SAXS to show that the tetramerization property can be transferred to other scFvs by characterizing a new construct harboring the same framework sequence and C-terminal tags, but in which the CDRs were substituted by those from a previously described anti-FLAG M2 scFv[19].

## Results

### The X-ray structure of scFv 2A2 reveals a cobalt-stabilized tetrameric assembly.
ScFv 2A2 crystallized in space group $I121$ in mother liquor containing 10 mM $CoCl_2$ and the structure was solved by molecular replacement at 2.5 Å resolution using a homology model of the scFv $V_H$ and $V_L$ domains as the search model (Table 1). The asymmetric unit is composed of 2 scFv molecules that form a tetramer with a twofold crystallographic axis, i.e. a dimer of dimers (Fig. 1b, Supplementary Data 1 and Supplementary Fig. 1). Each scFv monomer adopts the classical arrangement of the $V_H$ and $V_L$ domains, and further dimerizes through $a \sim 650$ Å$^2$ $V_L-V_L$ interface that is stabilized by 15 hydrogen bonds (Fig. 1d, f, g). This $V_L-V_L$ interface has previously been observed in the crystal packing of a number of scFv structures[18,20] and isolated $V_L$ domains[21], as can be seen from the structural alignment shown in Supplementary Fig. 1. A recent analysis of antibody fragment antigen-binding (Fab) X-ray structures in the PDB found that this $V_L-V_L$ interface was the 5th most common type of Fab-Fab interface with a prevalence of 3% (defined as VL-11 in ref. [22]). The most striking and unexpected feature of the structure is the presence of electron density for the C-terminal sortase recognition motif ($^{266}LPETG^{270}$), the 6XHis and the first Strep tag ($^{277}WSHPQFEK^{284}$ or only $^{277}WSHPQ^{281}$ in one of the two chains) of the Twin-strep tag (WSHPQFEK(G₃S)₃WSHPQFEK). These 16–19 residues form an extension that adopts a tetrameric arrangement stabilized by 8 $Co^{2+}$ ions (Figs. 1c and 2). The motif, named TetrHis (for Tetrameric His-Tag), packs against the scFv surface making hydrophobic contacts mainly involving framework residues Pro238, Phe241, Met243, Lys261, Glu263, and Leu266 and Trp277 from the motif (Fig. 1e). The tetrameric assembly contains two 6 residue-long β-strands related by the twofold symmetry axis, and the 8 $Co^{2+}$ ions coordinate 3–5 protein residues each, resulting in 4 unique metal binding sites (Fig. 2a). In site 1 and 2, $Co^{2+}$ ions are coordinated by residues belonging exclusively to the two 1st protein chains, thereby stabilizing the dimeric assembly. In contrast, the $Co^{2+}$ coordination at site 3 and 4 connects protein chains across the twofold crystallographic axis and thus stabilizes the tetramer (Fig. 2b). The $Co^{2+}$ ion of site 1 is coordinated by His279 (Strep tag) of the 1st chain and His273 and His275 (6XHis) of the 2nd chain (Fig. 2c). The coordination sphere is completed by 2 water molecules resulting in a distorted trigonal bipyramidal geometry. Site 2 adopts a distorted octahedral geometry with $Co^{2+}$ coordination by Glu268 (sortase), His273 and His275 (6XHis) of the 1st chain, His279 (Strep tag) of the 2nd chain, and a water molecule (Fig. 2d). Site 3 $Co^{2+}$ ion also exhibits an octahedral geometry and is coordinated by 5 protein residues (Fig. 2e): His276 from both the 1st and 2nd chains (6XHis), Glu268 (sortase), His272 and His274 (6XHis) from the 4th chain (symmetry equivalent of the 2nd chain). Finally, site 4 involves His272 and His274 from the 1st chain and His271 from the 4th chain (Fig. 2f). No additional water density is visible, resulting in a trigonal pyramidal geometry.

Overall, the sidechains of Glu268 from the sortase recognition motif, His279 from the strep tag, and all the histidines from the 6XHis (His271-276) are directly involved in binding to $Co^{2+}$ ions, with the exception of His271 from the 1st chain (Fig. 2b). Several important differences exist between the two non crystallographically related chains at the level of the tetramerization motif: (1) the β-strand from the 1st chain covers the $^{275}HHWSHP^{280}$ sequence and shares backbone-backbone hydrogen bonds with the 2nd chain β-strand which has the sequence $^{272}HHHHHW^{277}$ (Fig. 2); (2) while there is no electron density for residues after Gln281 of the 1st chain, in the 2nd chain residues $^{278}SHPQFEK^{284}$ interact with both the tetramerization

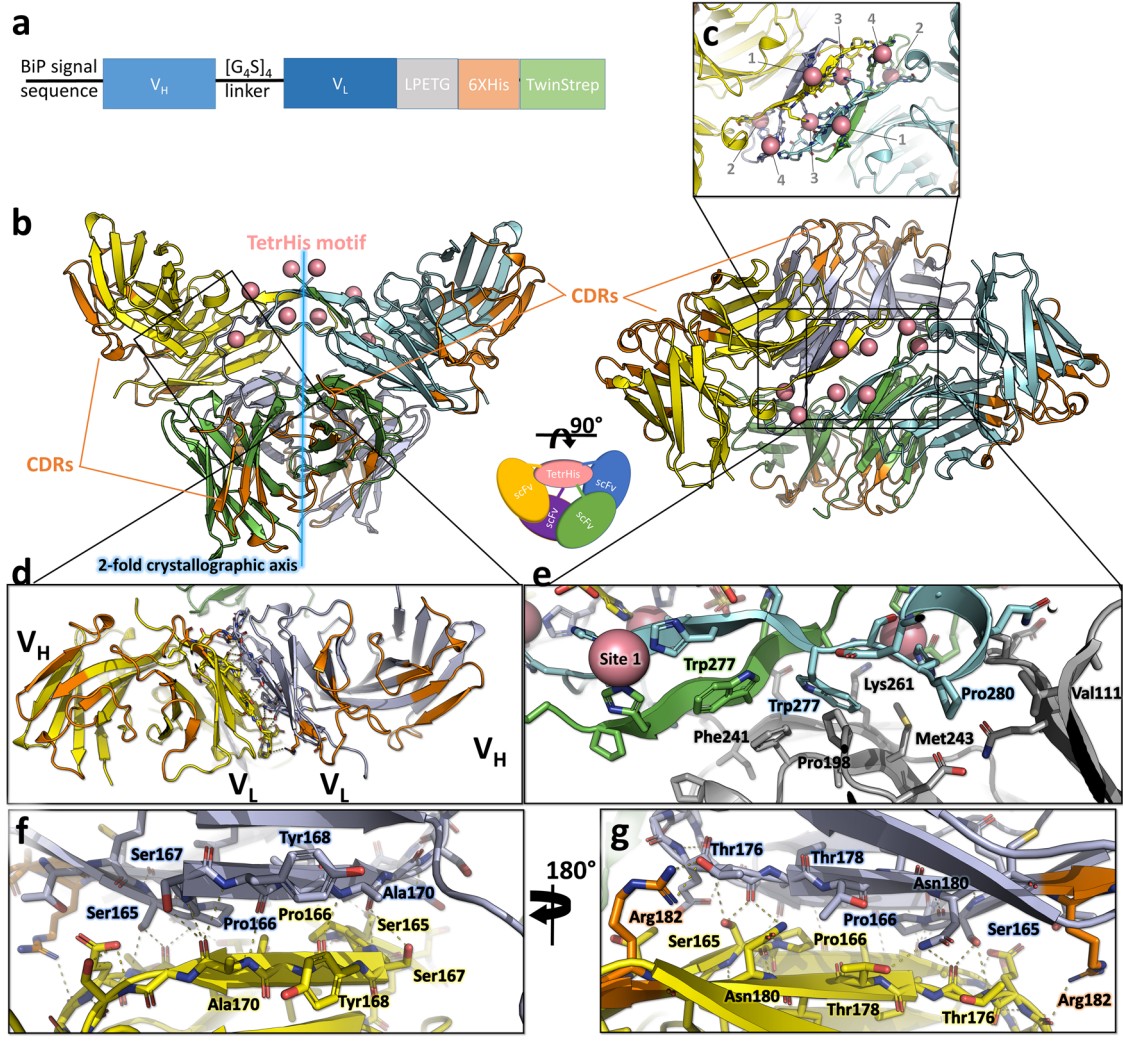

**Fig. 1 Crystal structure of the cobalt-bound scFv 2A2 tetramer. a** Protein construct used for crystallization of scFv 2A2. The expressed sequence contains a (cleaved) N-terminal BiP signal sequence and a C-terminal tag composed of a sortase motif (LPETG), a 6XHis and a twinstrep tag (WSHPQFEK[G$_3$S]$_3$WSHPQFEK). **b** Overall architecture of the tetramer in 2 views rotated by 90°. The protein chains are shown in cartoon representation and colored in green, cyan, yellow, and purple. The complementarity determining regions (CDRs) are colored in orange. The 8 cobalt ions are shown as salmon-colored spheres. **c** Close up view of the cobalt-bound TetrHis motif showing the 4 types of metal binding sites related by the twofold symmetry axis. **d** Close up view of the dimeric V$_L$−V$_L$ interface (defined as VL-11 in ref. [22]). **e** Hydrophobic interactions stabilizing the interface between the TetrHis motif and the scFv V$_H$ and V$_L$ domains. Framework residues belonging to the V$_L$ domain are shown in gray. **f, g** Details of the V$_L$−V$_L$ interface interactions, in 2 views rotated by 180°.

motif and residues from the scFv framework, forming an α-helical turn (Figs. 1c, e and 2a). From both chains, Trp277 play an important stabilizing role in the complex, creating a hydrophobic core between the tetramerization motif and the V$_L$ domain of the 2nd chain, however Trp277 from the 1st chain remains largely solvent-exposed while Trp277 from the 2nd chain is buried between the scFv and the TetrHis motif (Fig. 1e).

**Analysis of metal-bound homopolymeric histidine tracts in the PDB shows that the TetrHis motif has no known homologs.** In order to determine whether similar tetramerization motifs involving metal-interacting polyhistidine tags existed in the PDB, we performed a survey of available experimental structures using the PDB structure motif search tool. We used four consecutive histidines from the 6XHis of entry 2JSN[23] as a search motif and an arbitrarily high RMSD cut-off in order to avoid excluding structures in which histidines adopt a different conformation. Out of the 81 entries that contained at least four consecutive visible

histidines and any bound zinc, copper, nickel, cobalt, cadmium, or iron, we identified 63 structures in which at least one metal ion interacts directly with the 6XHis motif. We curated this initial dataset by removing redundancy, grouping highly similar structures together and keeping only a single representative, which yielded 27 unique X-ray crystallographic structures (Supplementary Table 1). We further omitted structures in which less than two histidines from the 6XHis were involved in metal coordination. Basic statistics regarding the nature, number of bound metal and their interactions with the protein are summarized in Fig. 3. Most of these structures contain bound zinc (12) or nickel (9), and the number of metal ions is generally four or less (23 out of 27 structures). These metals usually stabilize the crystal packing by connecting two protein chains (20 out of 27 structures). We found only five occurrences of linear metal-binding motifs (i.e. short amino acid sequences typically ranging from 3 to 10 residues in length) out of the 27 structures, all of which involve one or two bound nickel ions, stabilizing either a dimeric or trimeric assembly (Fig. 3e–i). In all five cases, it is

**Table 1 Data collection and refinement statistics (molecular replacement).**

|  | scFv 2A2 |
| --- | --- |
| Data collection |  |
| Space group | I 1 2 1 |
| Cell dimensions |  |
| $a, b, c$ (Å) | 97.50, 81.82, 111.04 |
| $\alpha, \beta, \gamma$ (°) | 90.00, 111.14, 90.00 |
| Resolution (Å) | 45.47–2.50 (2.60–2.50) |
| $R_{merge}$ | 0.213 (1.293)[a] |
| $CC_{1/2}$ | 0.989 (0.491) |
| $I / \sigma I$ | 5.6 (1.1) |
| Completeness (%) | 99.2 (96.8) |
| Redundancy | 6.9 (6.2) |
| Refinement |  |
| Resolution (Å) | 45.47–2.50 |
| No. reflections | 28,159 (1387)[b] |
| $R_{work}/R_{free}$ | 19.02/24.83 |
| No. atoms |  |
| Protein | 3847 |
| Cobalt | 4 |
| Water | 196 |
| B-factors |  |
| Protein | 47.8 |
| Cobalt | 60.4 |
| Water | 46.0 |
| R.m.s. deviations |  |
| Bond lengths (Å) | 0.008 |
| Bond angles (°) | 1.318 |
| Ramachandran statistics |  |
| Favored | 97.31 % |
| Outliers | 0.41 % |

[a]Values in parentheses are for highest-resolution shell.
[b]Values in parentheses are for the test set.

unknown whether the crystallographically observed assembly can also form in solution in the presence of $Ni^{2+}$ ions, however entries 3CGM and 4ODP are crystal structures of SlyD from Thermus thermophiles, a protein well known for its metallo-chaperone activity[24–26]. Taken together, these analyses indicate that no linear metal-dependent polyhistidine tetramerization motif exists in the PDB, making TetrHis a novel sequence motif.

**DLS analyses show that the size of scFv 2A2 increases as a function of cobalt concentration and that the changes are mediated by the TetrHis motif.** In order to assess the ability of cobalt to induce changes in the conformation or oligomeric state of scFv 2A2 in solution, we performed a cobalt titration by DLS (Fig. 4). In addition, we used an anti-ADIPOR scFv bearing a C-terminal twin strep tag[18] as a negative control to determine whether the observed changes are indeed related to the presence of the TetrHis motif. We found that addition of 500 μM to 5 mM of cobalt ions led to a significant shift in the size distribution and calculated hydrodynamic radius ($R_h$) of scFv 2A2 (Fig. 4 and Supplementary Fig. 2). The estimated $R_h$ increased from ~3.3 nm in the absence of cobalt to ~4.8 nm at 10 mM $Co^{2+}$, indicating the formation of larger species. These values compare well with the ones calculated from structural models of monomers (3.2 nm) and tetramers (4.4 nm) extracted from the x-ray structure, after addition of missing residues (interdomain linker and second strep tag). On the contrary, cobalt seemed to induce a compaction of the control scFv from a $R_h$ value of 5.6 nm down to 3.4 nm, with a concomitant decrease in the width of the size distribution (Supplementary Fig. 2). Although the initial compaction from 5.5 to

4.0 nm in 200 μM $CoCl_2$ was unexpected, all the data measured in the presence of cobalt for this scFv are consistent with a monomeric state. These data demonstrate that the TetrHis motif can mediate cobalt–dependent changes in the conformation and/or oligomeric state of scFv 2A2 in solution, which are compatible with tetramer formation.

**SAXS analysis indicates metal-induced changes in the structure and oligomeric state of scFv 2A2.** Intrigued by the peculiar tetrameric architecture of scFv 2A2 observed in the crystal structure, we turned to SAXS in order to assess the structure of the scFv directly in solution. SAXS profiles were measured at three or five different protein concentrations in gel filtration buffer or in the presence of additives: 5 mM EDTA, 5 mM $NiSO_4$, or 5 mM $CoCl_2$ (Fig. 5a and Table 2). We also attempted to measure data in the presence of 5 mM $ZnCl_2$, but these showed severe signs of aggregation. The measured radius of gyration (Rg) in the presence of EDTA increased from 24 Å at 2 mg/ml of protein to 29 Å at 8 mg/ml, while the Rg values in regular buffer were comprised between 25 and 32 Å, at 0.5 and 8 mg/ml of protein, respectively. This concentration dependence of the Rg suggested possible interparticle attraction, however the measured values were roughly consistent with the Rg calculated from monomers extracted from the crystal structure, particularly at low protein concentration (theoretical Rg ≈ 20–28 Å dependent on whether the missing residues have been added or not, and their respective conformations). In the presence of $Co^{2+}$ or $Ni^{2+}$ ions, the measured SAXS profiles showed a noticeable change of slope between $Q ≈ 0.1$–$0.2$ Å$^{-1}$, particularly visible at high protein concentration (Fig. 5a), suggesting significant hollowness of the scattering object. This feature was consistent with the presence of a large void within the tetrameric scFv structure (Fig. 1b), with the presence of a channel of 10–20 Å width in between the TetrHis motif and scFv domains. Similar to the data measured in the absence of added metal ions, the measured Rg showed important protein concentration-dependent variations with values of 36–60 Å in the presence of $Ni^{2+}$, and 34–50 Å in the presence of $Co^{2+}$ (Table 2). The 34 to 36 Å Rg values obtained at low protein concentration were also roughly consistent with the Rg calculated from the crystallographic tetramer (32–36 Å after addition of the scFv linker and missing residues at the C-terminus). Comparison of the Kratky plots from SAXS data measured at protein concentrations of 2 mg/ml in the presence of EDTA versus $Co^{2+}$ ions suggests that the metal ions induce a transition from a mostly globular to a multidomain (or oligomeric) protein (Fig. 5b). The pair-distance distribution functions p(r) calculated from the SAXS profiles measured in the presence or absence of 5 mM $Co^{2+}$ at different protein concentrations are shown in Fig. 5c. The addition of cobalt leads to a pronounced shift of the p(r) towards larger interatomic distances, which further increases with protein concentration. At 1–2 mg/ml of protein, the presence of 2 overlapping peaks in the distribution is clearly visible, while higher protein concentrations seem to induce the formation of even larger species.

Because the strong concentration dependence of the measured Rg values made analysis of solution data less straightforward, we next measured SEC-SAXS data with or without addition of $Co^{2+}$ ions in the SEC buffer. We hypothesized that the size separation of the SEC would remove potential interparticle interference or signal contributions from small amounts of higher order oligomers or aggregates, and thus provide a useful comparison with the solution-based data. ScFv 2A2 eluted from the SEC as a single, symmetric peak as can be seen from the average intensity and $UV_{280}$ profiles (Fig. 5c and Supplementary Fig. 3). The measured Rg was mostly constant across the peak with a slight

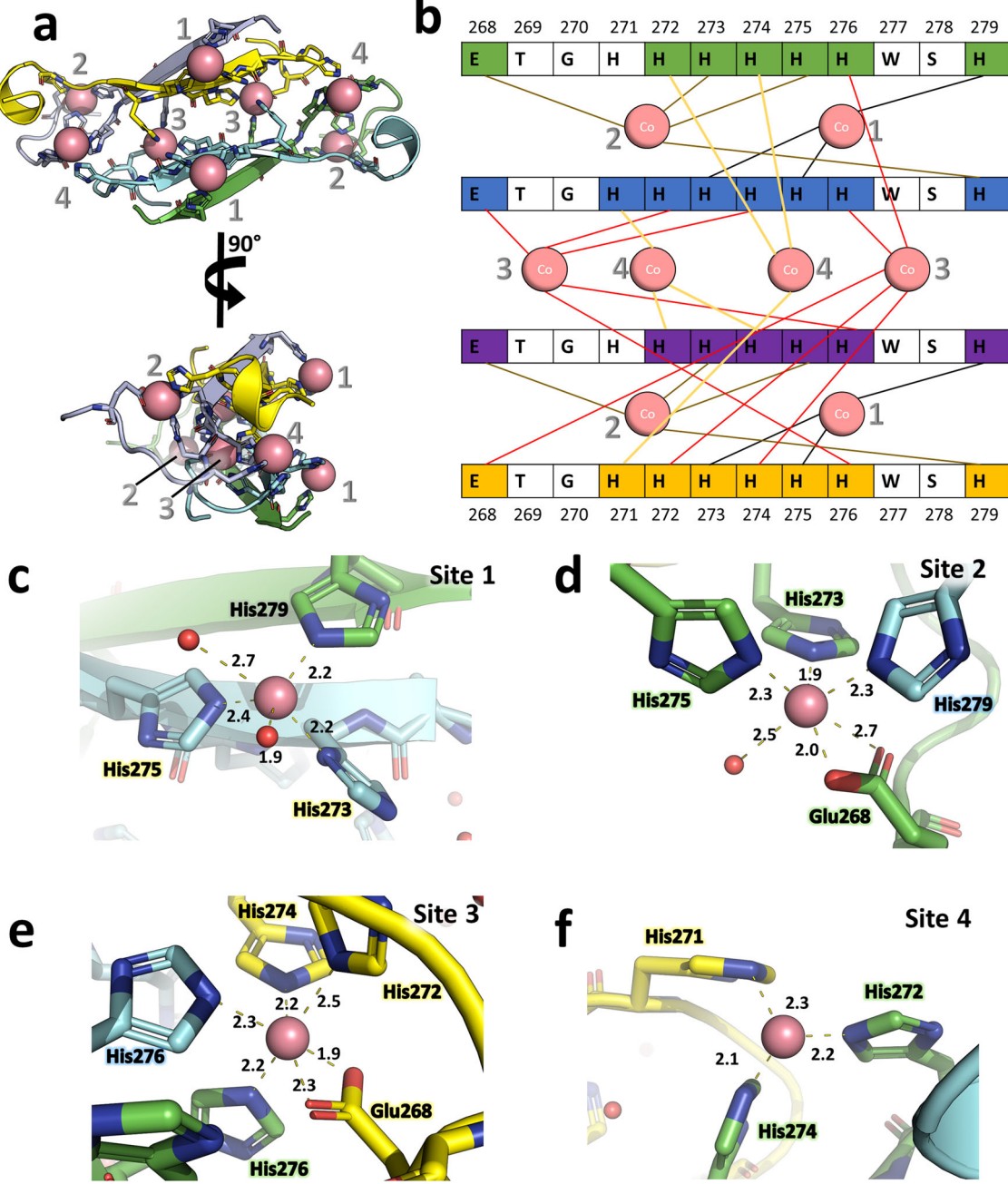

**Fig. 2 Architecture of the cobalt-bound tetrameric polyhistidine (TetrHis) motif. a** Close up of the TetrHis motif, in two views rotated by 90°. The protein chains are shown in cartoon representation and the color code is the same as in Fig. 1. **b** Schematic representation of the metal-protein contacts within the tetrameric motif. **c–f** Close ups of the 4 cobalt binding sites showing the coordination geometries and distances.

downward trend in the second half of the peak and an average value of 26.5 Å, in agreement with the Rg value expected for a monomeric species (Table 2 and Fig. 5d). The estimated molecular weight (MW) was also mostly constant ranging from 28 to 33 kDa (Supplementary Fig. 3b) in agreement with the theoretical MW of 30.5 kDa. In contrast, the profile measured in the presence of 5 mM CoCl$_2$ displayed a wider, assymmetric peak that was slightly shifted towards lower elution volumes (Fig. 5d). The measured Rg showed important variations across the peak with values ranging from ~ 25 to 34 Å (Fig. 5d) and the estimated MW was comprised between 30 and 60 kDa (Supplementary Fig. 3b). All these observations indicate cobalt-induced heterogeneity in the scFv oligomeric state.

**Molecular dynamics simulations suggest significant scFv interdomain flexibility.** In order to analyze the conformational landscape of scFv 2A2 using the measured SAXS data, we generated the ensembles of conformers required for SAXS-based ensemble optimization using classical explicit solvent MD simulations. We ran two independent trajectories of the monomer (Fig. 6a and Supplementary Fig. 4a). In the first simulation, the 39 residue-long C-terminal extension composed of the LPETG, 6XHis, and TwinStrep tag quickly packed against the surface of the scFv and remained stable for the remainder of the simulation. In the second simulation, the C-terminal extension remained highly flexible and we observed complete dissociation of the V$_H$ and V$_L$ domains in the 2nd half of the trajectory (Fig. 6a). This spontaneous transition

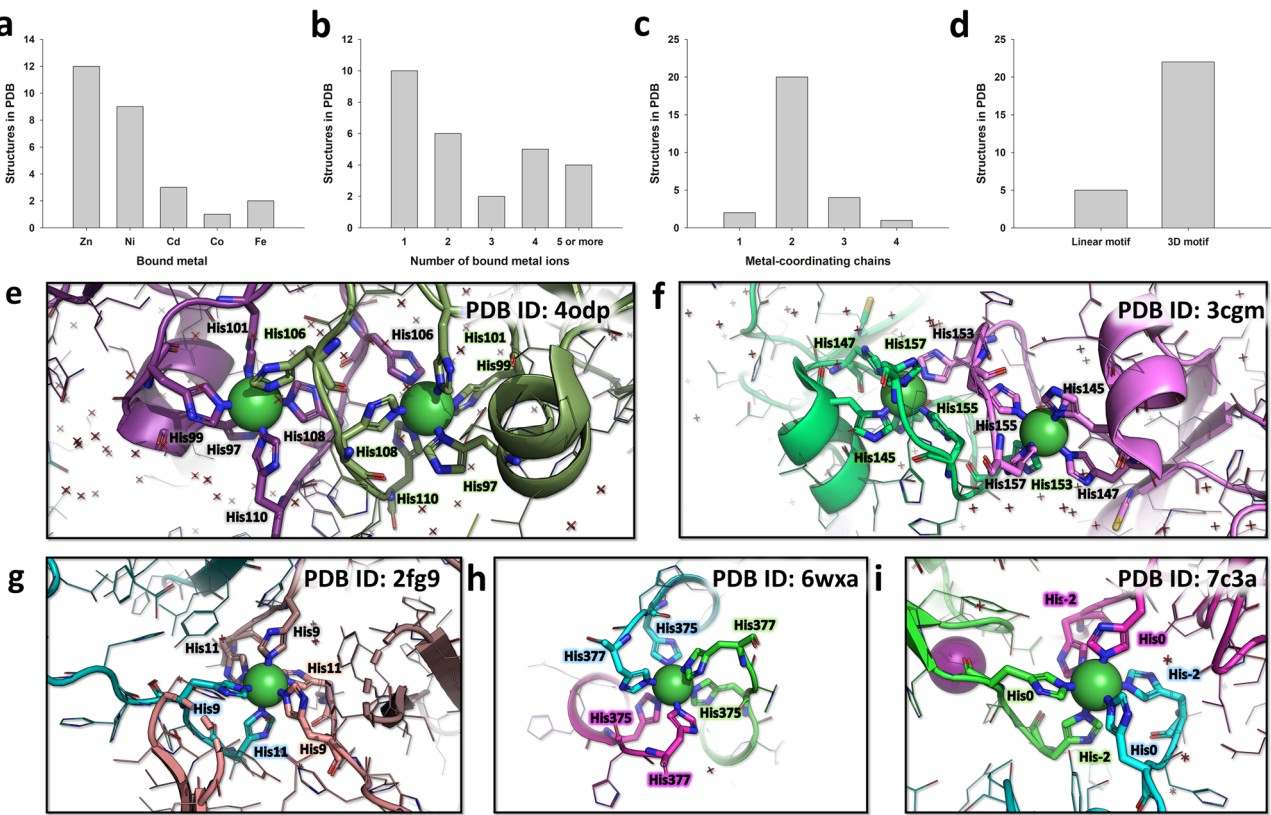

**Fig. 3 Survey of metal-bound polyhistidine tags in the PDB. a–d** Basic statistics extracted from the 27 unique PDB structures containing at least 4 consecutive histidines and a bound metal. The structures are classified according to the nature of the bound metal (**a**), the number of bound metal ions in the assembly (**b**), the number of protein chains that contribute coordinating residue sidechains (**c**), and whether the assembly involves a short linear sequence motif or a three-dimensional motif (with residues located far away within the protein sequence) (**d**). **e, f** Nickel-dependent dimeric assemblies of SlyD from Thermus thermophiles, observed in PDB entries 4odp and 3cgm. 3 His residues located at the C-terminus of the construct are involved in metal coordination, along with 2 His from a 6XHis tag located 3 residues downstream, and 1 His from the 6XHis tag of a symmetric-related molecule. **g–i** Nickel-dependent trimeric assemblies observed in PDB entries 2fg9, 6wxa, and 7c3a. In each case, 3 × 2 His residues from 6XHis tags assemble to form a octahedral coordination sphere around a single nickel ion.

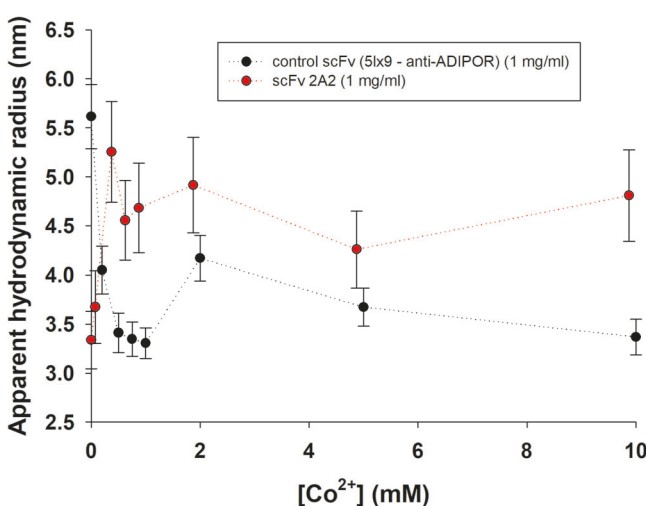

**Fig. 4 Cobalt titrations of scFv 2A2 and anti-ADIPOR scFv by dynamic light scattering (DLS) at 20 °C in 50 mM HEPES pH 7.5, 150 mM NaCl using a protein concentration of 1 mg/ml.** Each point corresponds to the average of 3 independent measurements, with standard error shown as a bar. The apparent hydrodynamic radii were calculated from the intensity autocorrelations using the Stokes–Einstein equation.

to an open conformation suggested limited stability of the scFv $V_H-V_L$ interface, which is a relatively frequent property in scFvs that have not been engineered for stability[27,28]. We also ran multiple independent MD trajectories of the tetrameric scFv complex (Fig. 6b and Supplementary Fig. 4b) in which the cobalt ions were replaced by zinc ions (as parameters for $Co^{2+}$ are not available in standard MD force fields). In all trajectories but one, we applied distance restraints to the metal coordination centers in order to maintain the integrity of the crystallographically observed tetramerization motif, resulting in stable tetramers. In the absence of restraints, we observed that zinc ions tended to dissociate over time, leading to partially dissociated tetrameric states at longer timescales (Fig. 6b). We then extracted all monomeric and tetrameric scFv conformers observed in the MD trajectories to create a large ensemble of models for SAXS-based ensemble optimization.

**Optimization of MDS-derived ensembles against SAXS data reveals the conformational landscape of scFv 2A2 in the absence and presence of cobalt(II) ions.** The ensemble optimization method (EOM) uses a genetic algorithm to optimize a small (usually 5–20) ensemble of models that fits the experimental SAXS data from a larger pool of structural models (>1000)[29]. This method was used to fit SAXS curves extracted from both the solution-based and SEC-SAXS experiments using the pool of MD models of scFv 2A2 (Fig. 7). The data measured in the absence of cobalt was generally well fitted using the monomeric ensemble of models with

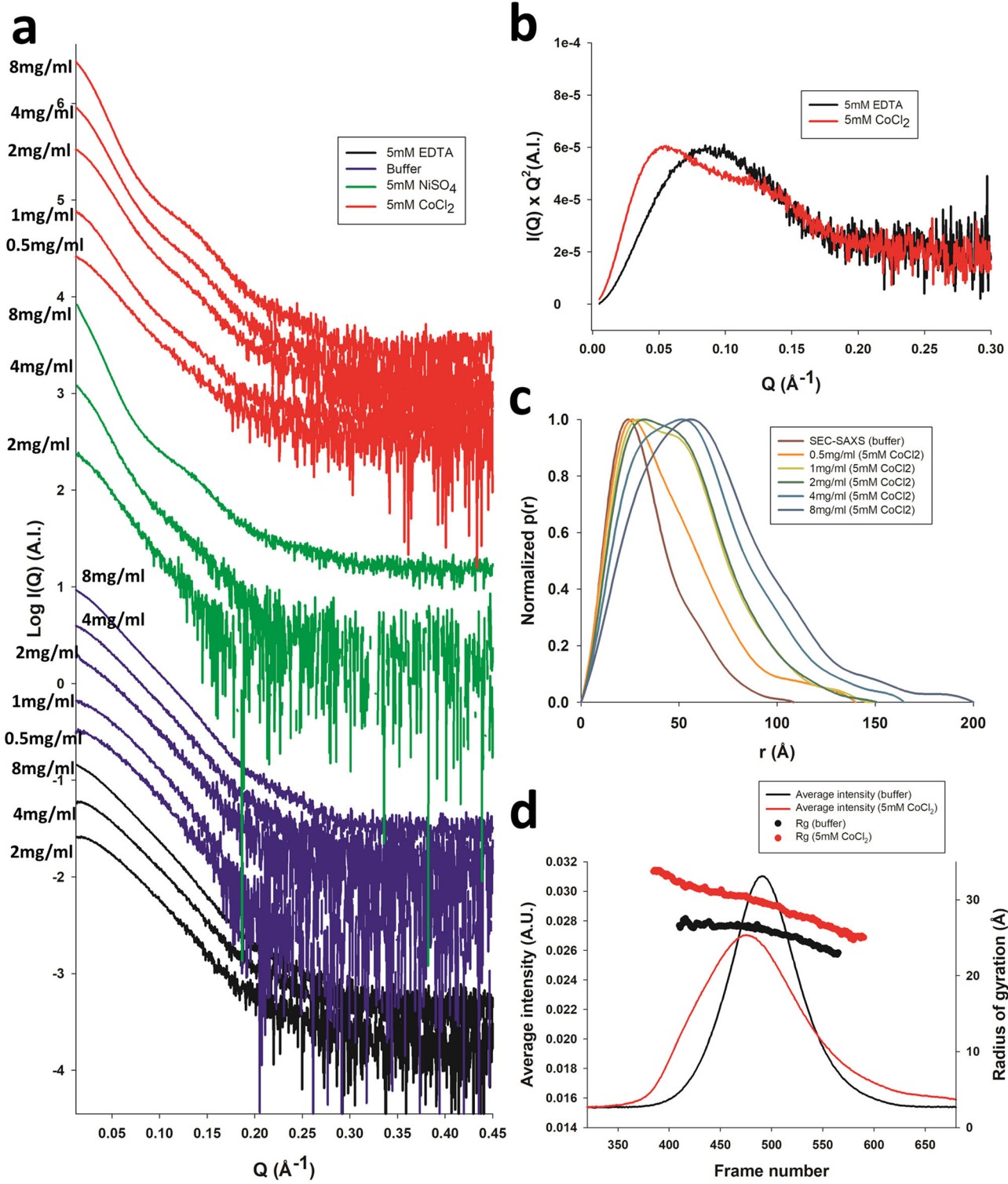

**Fig. 5 SAXS characterization of scFv 2A2. a** SAXS profiles of scFv 2A2 in the presence of 5 mM EDTA (black curves), in regular buffer (50 mM HEPES pH 7.5 and 150 mM NaCl, blue curves), or in the presence of 5 mM NiSO$_4$ or 5 mM CoCl$_2$ (green and red curves, respectively). The data were measured at increasing protein concentrations of 2.1, 4.2, and 8.5 mg/ml for EDTA and NiSO$_4$, while additional curves were measured at 0.5 and 1mg/ml in regular buffer and CoCl$_2$ conditions. **b** Kratky plots the EDTA and CoCl$_2$ data using the 2.1 mg/ml SAXS profiles. **c** Pair-distance distribution functions p(r) calculated from the SAXS profiles measured in the presence or absence of 5 mM Co$^{2+}$ at different protein concentrations. The SAXS curves extracted from the SEC-SAXS profile is used as the cobalt-free reference. **d** SEC-SAXS profiles of scFv 2A2 in the presence (red) or absence of 5 mM CoCl$_2$ (black curve), and analysis of Radius of gyration versus frame number.

**Table 2 SAXS-derived parameters.**

| | Buffer conditions | $c$ (mg/ml) | MW (kDa) | $R_g$ (Å) | $\chi_{EOM}$ |
|---|---|---|---|---|---|
| ScFv 2A2 SAXS | | | | | |
| | Buffer A[a] | 8.5 | 47.6 | 32.3 ± 0.2 | 1.604 |
| | Buffer A | 4.2 | 40.6 | 30.0 ± 0.2 | 1.391 |
| | Buffer A | 2.1 | 35.8 | 27.6 ± 0.3 | 1.079 |
| | Buffer A | 1.0 | 35.6 | 25.5 ± 0.2 | 0.943 |
| | Buffer A | 0.5 | 29.9 | 24.7 ± 0.3 | 0.901 |
| | Buffer A + 5 mM EDTA | 8.5 | 39.7 | 29.0 ± 0.3 | 2.185 |
| | Buffer A + 5 mM EDTA | 4.2 | 29.4 | 25.6 ± 0.2 | 1.110 |
| | Buffer A + 5 mM EDTA | 2.1 | 26.1 | 24.1 ± 0.2 | 1.039 |
| | Buffer A + 5 mM $CoCl_2$ | 8.5 | 172.1 | 49.9 ± 0.2 | 10.279 |
| | Buffer A + 5 mM $CoCl_2$ | 4.2 | 122.8 | 44.2 ± 0.2 | 4.770 |
| | Buffer A + 5 mM $CoCl_2$ | 2.1 | 72.3 | 38.1 ± 0.2 | 1.695 |
| | Buffer A + 5 mM $CoCl_2$ | 1.0 | 69.9 | 38.4 ± 0.2 | 1.274 |
| | Buffer A + 5 mM $CoCl_2$ | 0.5 | 58.1 | 33.9 ± 0.2 | 0.927 |
| | Buffer A + 5 mM $NiSO_4$ | 8.5 | 248.5 | 60.2 ± 0.5 | N.D. |
| | Buffer A + 5 mM $NiSO_4$ | 4.2 | 126.4 | 45.2 ± 0.7 | 2.233 |
| | Buffer A + 5 mM $NiSO_4$ | 2.1 | 75.4 | 36.3 ± 0.8 | 1.034 |
| SEC-SAXS | | | | | |
| Frame 464-524 | Buffer A | 0.85[b] | 34.7 | 26.5 ± 0.1 | 1.083 |
| Frame 390-455 | Buffer A + 5 mM $CoCl_2$ | 0.85[b] | 55.4 | 31.7 ± 0.1 | 1.031 |
| ScFv748 SAXS | Buffer A | 0.5 | 32.6 | 25.1 ± 0.3 | 0.916 |
| | Buffer A + 5 mM $CoCl_2$ | 0.5 | 47.1 | 34.9 ± 0.4 | 0.946 |

[a]Buffer A: 50 mM HEPES pH 7.5 150 mM NaCl.
[b]Concentration of the sample that was injected on the SEC assuming a 10-fold dilution factor.

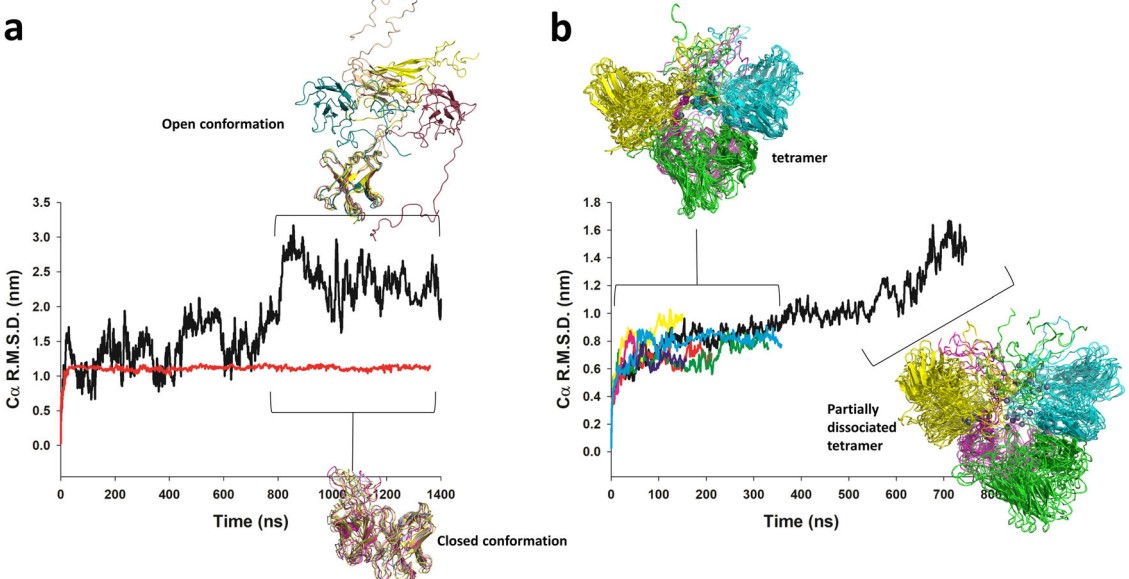

**Fig. 6 Molecular dynamics simulations analysis of scFv 2A2 in its monomeric and tetrameric forms. a** Cα root mean square deviation (R.M.S.D) as a function of simulation time for two independent MD trajectories of the scFv monomer. Five representative conformers of the open and closed states sampled during the simulations are superimposed and shown as cartoon. **b** Cα R.M.S.D as a function of simulation time for the MD trajectories of the scFv tetramer. Representative conformers along the trajectories are shown in cartoon representation and colored by chain. The longer trajectory (black curve) was run without distance restraints on metal binding sites, which resulted in the sampling of partially dissociated tetrameric states.

$\chi_{EOM}$ values of 1.0–1.6, except for a slight worsening of the quality of fit at high protein concentration in the presence of EDTA ($\chi_{EOM} = 2.185$; Table 2). The Rg distribution obtained from analysis of the SEC-SAXS data is shown in Fig. 7a. The distribution shows a single peak centered on the experimental Rg value of 26.5 Å for models of the selected ensemble. The ensemble is composed of 80–85% of closed conformers and 15–20% of open conformers (Fig. 7e), and the C-terminal extension behaves as an intrinsically disordered tail adopting mostly extended conformations (Fig. 7f).

For the SEC-SAXS data measured in the presence of cobalt ions, a SAXS curve was extracted from the first part of the asymmetric elution peak (frames 390-455, see Fig. 5d). The data could not be adequately fitted using the monomeric scFv ensemble ($\chi_{EOM} = 4.64$) and required a pool ensemble containing both monomeric and tetrameric forms (Fig. 6b) to reach a satisfactory $\chi_{EOM}$ value of 1.03 (Fig. 7b). We also systematically fitted all solution-based SAXS profiles against this monomer-tetramer ensemble and found that good agreement with the

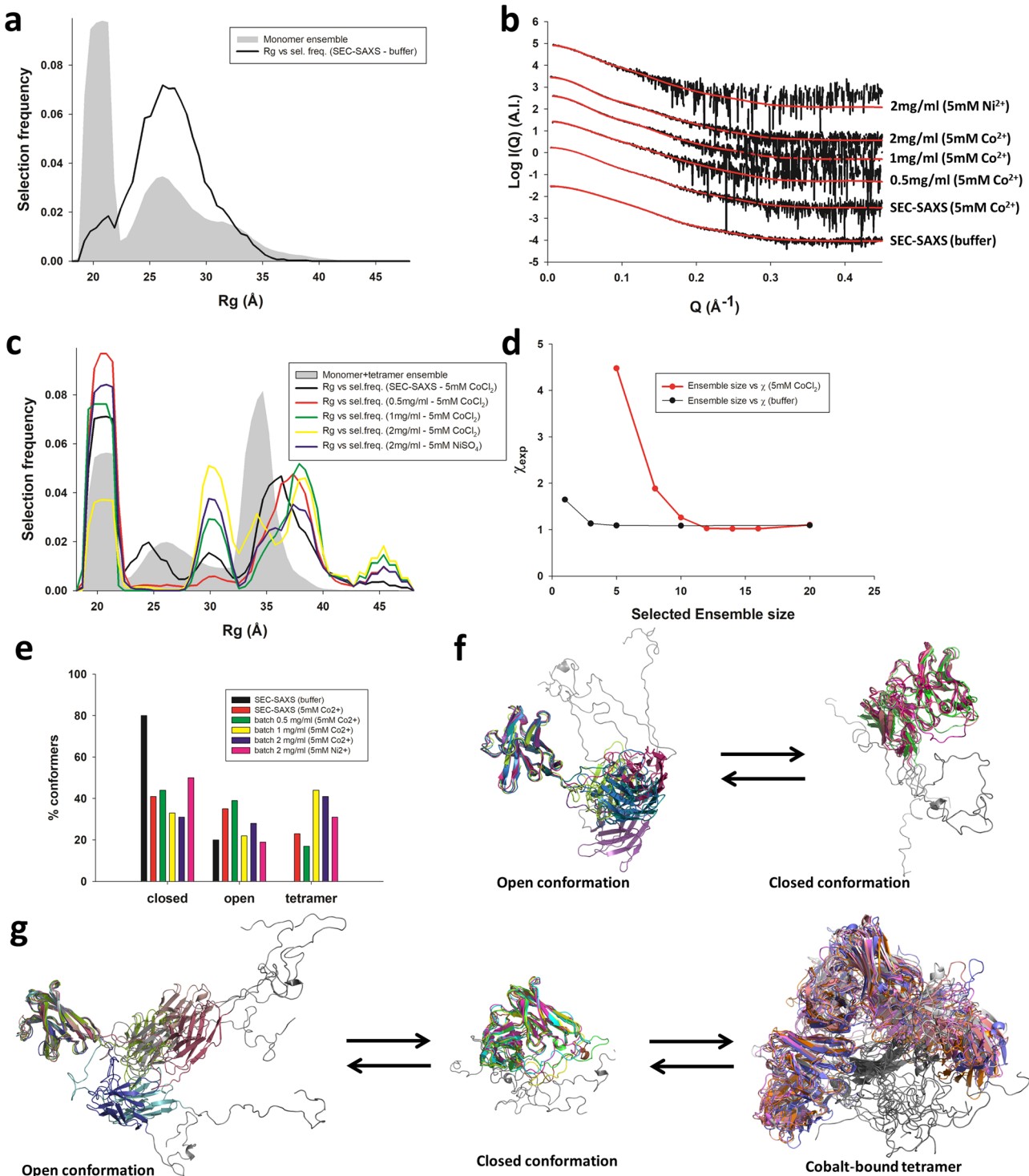

**Fig. 7 Ensemble optimization (EOM) analysis of the SAXS curves extracted from the SEC-SAXS profiles of scFv 2A2. a** Radius of gyration distributions for the MD-generated ensemble of monomeric models (gray area) and for the optimized ensemble (OE) that fits the SEC-SAXS data in the absence of cobalt (black curve). **b** Fitted SAXS profiles of scFv 2A2 measured in the presence or absence of 5 mM $NiSO_4$ or $CoCl_2$ in solution or in SEC-SAXS. The protein concentration for each curve is indicated on the figure panel. Experimental scattering curves are shown as black lines with OEs fits shown as red lines. **c** Radius of gyration distributions for the MD-generated ensemble of monomeric and tetrameric models (black area) and for the OEs that fits the solution-based or SEC-SAXS data (curves) in the presence of 5 mM $CoCl_2$ or 5 mM $NiSO_4$. **d** EOM Goodness-of-fit $\chi_{exp}$ as a function of the number of models in the OEs, in the presence (red) or absence (black) of 5 mM $CoCl_2$. **e** Histograms showing the percentage of tetramers, open and closed state monomers observed in the best fitting OEs for different experimental conditions. **f** Conformational equilibrium of monomeric scFv 2A2 in 50 mM HEPES pH 7.5 and 150 mM NaCl. For each state, 5 representative models extracted from the OEs are superimposed and shown as cartoon. The disordered C-terminal tail corresponding to the sortase-His6-Twinstrep sequence is colored in gray. **g** Conformational equilibrium of scFv 2A2 in the presence of 5 mM $CoCl_2$. For each state, 5 representative models extracted from the OEs are superimposed and shown as cartoon. The sortase-His6-Twinstrep tail encompassing the TetrHis motif is colored in gray.

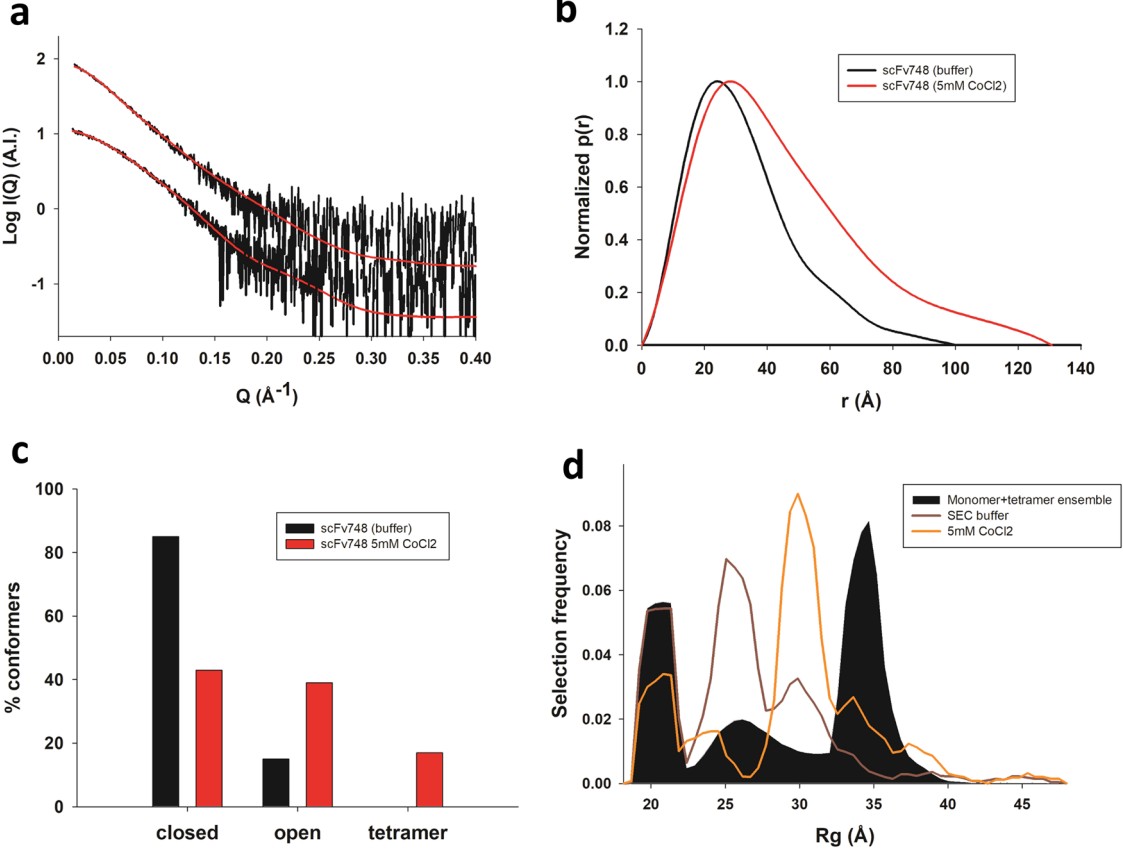

**Fig. 8 SAXS characterization and ensemble optimization of scFv 748. a** Fitted SAXS profiles of scFv 748 measured in the presence or absence of 5 mM CoCl$_2$ in solution at 0.5 mg/ml protein concentration. **b** Pair-distance distribution functions p(r) calculated from the SAXS profiles measured in the presence or absence of 5 mM CoCl$_2$. **c** Proportion of open and closed states monomers and tetramers in the OEs in the presence or absence of 5 mM CoCl$_2$. **d** Radius of gyration distributions for the MD-generated ensemble of monomeric and tetrameric models (black area) and for the OEs that fit the solution SAXS data (brown and orange curves) in the presence or absence of 5 mM CoCl$_2$.

experimental data could be obtained for protein concentrations of up to 2 mg/ml (Fig. 7b and Table 2). The Rg distributions reveal a complex landscape induced by the cobalt or nickel ions (Fig. 7c). The increased complexity of the scFv conformational landscape is also apparent in the larger number of models required to fit the SAXS data (~ 15 versus 5 - as illustrated for the SEC-SAXS measurements in Fig. 7d. Analysis of the selected ensembles indicate a dynamic equilibrium between tetramers, closed state and open state monomers (Fig. 7e, g). In 5 mM Co$^{2+}$, the population of tetramers increases from roughly 20–40% when protein concentration goes from 0.5 to 1 or 2 mg/ml (Fig. 7e), whereas about 25% of tetramers are present in the SEC-SAXS data, which is consistent with the ~ 10-fold dilution that occurs on the SEC. At 2 mg/ml of protein in the presence of 5 mM Ni$^{2+}$, we also observed a comparable proportion of tetrameric states of ~ 30%, suggesting that both metals are able to induce tetramer formation in a similar way. Interestingly, the selected closed state monomers are much more compact than those selected in the absence of cobalt ions (Fig. 7a vs Fig. 7c, and Fig. 7f vs Fig. 7g), suggesting cobalt-induced structural preorganization of the C-terminal extension in the monomeric state. In addition, the overall ratio of open to close monomeric states is higher in the presence of cobalt, which is consistent with a depletion of the close state conformers through tetramerization. Taken together, these results indicate that the presence of cobalt ions remodels the conformational landscape of scFv 2A2 by inducing tetramerization and affecting its intrinsic open-close state equilibrium in the monomeric form.

**SAXS characterization of scFv 748 demonstrates transferability of the TetrHis motif to other scFvs.** Based on the crystal structure of the cobalt-bound tetramer, all of the structural elements required for tetramerization (i.e. TetrHis motif, V$_L$- V$_L$ interface and the scFv region onto which the TetrHis motif packs) are located outside of the CDR loops. An exception to this is the 1$^{st}$ residue of CDR1 of the V$_L$ domain (Arg182), which contributes 1 hydrogen bond at the periphery of the V$_L$- V$_L$ interface. In order to test whether other scFvs can also use the TetrHis motif for tetramerization, we designed a new scFv sequence by replacing the CDR loops in scFv 2A2 by those from an anti-FLAG M2 Fab[19], for which a crystal structure is available (PDB ID:2g60). The protein (named scFv 748), which shares 84.5% sequence identity with the original scFv, was produced with a pelB leader sequence and purified from e.coli periplasm (see methods). Although the purification yield obtained by this approach was limited, most probably due to lack of optimization of the framework sequence for bacterial expression, we could purify enough scFv 748 to test the effect of 5 mM of cobalt ions onto the measured SAXS profiles at a protein concentration of 0.5 mg/ml (Fig. 8a). The changes in the p(r) functions recapitulated the observations made in the same buffer and at the same protein concentration for scFv 2A2 (Figs. 8b and 5c). The curves could be well fitted by EOM using the monomer-tetramer ensemble, and yielded similar proportions of tetramers and monomers, as well as similar Rg distributions (Fig. 8c, d). Taken together, these results indicate that TetrHis – cobalt mediated tetramerization can be applied to various scFvs, provided that the same framework sequence and C-terminal tags are used.

## Discussion

In this study we characterized the structure and dynamics of scFv 2A2, uncovering a metal-dependent tetramerization motif containing a 6XHis sequence, which we called TetrHis. The motif possesses a unique β-stranded architecture and harbors 8 metal binding sites clustered within a small region of space, with inter-site distances of 10–30 Å. This discovery was made possible thanks to the serendipitous combination of (1) the right C-terminal tags; (2) an antibody framework sequence capable of stabilizing the motif; (3) successful crystallization in the presence of a high concentration of cobalt ions. The tetrameric assembly was further stabilized by the formation of VL-11 β-sheet dimers at the $V_L-V_L$ interfaces, a common type of weak homotypic antibody interface[22].

Using the PDB structure motif search tool, we found metal-coordinated polyhistidine sequences to be relatively rare in experimental structures. In most cases, a 6X His sequence was involved in crystal packing, stabilizing a dimeric state through metal coordination in conjunction with other residues located far away in the protein sequence. We were unable to identify any motif structurally homologous to TetrHis in the PDB, nor did we find any examples of polyhistidine motifs stabilizing tetrameric states.

Our analysis of DLS, solution, and SEC-SAXS data unambiguously showed that the existence of the TetrHis motif is not limited to the crystal, and that the tetrameric complex can also form in solution. In regular buffer conditions, we found that scFv 2A2 mostly adopted the classical scFv fold, but was in equilibrium with a minor population of open states resulting from the dissociation of $V_H$ and $V_L$ domains. This appears to be a frequent property of scFvs that have not been engineered for stability[27,28], although to the best of our knowledge, such equilibrium has not previously been described in atomistic details by SAXS-based ensemble analysis. In addition, we observed that the C-terminal extension behaved as a typical intrinsically disordered region.

In contrast, when $Co^{2+}$ ions were added to the buffer, we observed structural changes in the intrinsically disordered C-terminal extension with the appearance of a compact close state population in which the tags pack onto the scFv surface. The formation of tetramers, stabilized by the metal-bound TetrHis motif, was found to increase with protein concentration in the 0.5 to 1 mg/ml range, and plateau at 40 % between 1 and 2 mg/ml. For data measured at higher protein concentrations, we found that the quality of fit deteriorated, probably due to the formation of a small population of higher order oligomers and/or inter-particle interference. However, the analysis of p(r) distributions, and even the composition of the (poorly fitting) optimized ensembles, suggests that the population of tetramers keeps increasing at these concentrations. Furthermore, the fact that the protein crystallized as a tetramer at 4–8 mg/ml in 5–10 mM $CoCl_2$ seems to indicate that this should be the dominant scFv conformation in these conditions. Taken together, the SAXS analysis and DLS measurements suggest that tetramer formation is favored at high protein concentrations with cobalt concentrations in the mM range (e.g., 2–10 mM).

Because the atomic contacts between the TetrHis motif and the scFv tetramer are located in the scFv framework regions rather than the CDRs, it is reasonable to assume that the metal-dependent tetramerization property could be transferred to other scFvs provided the same antibody framework and C-terminal tags are used. We tested this hypothesis by producing scFv 748, in which the CDRs present in scFv 2A2 were replaced by those from a previously crystallized anti-FLAG M2 Fab[19]. Our SAXS data analysis showed that the new scFv behaved very similarly to the original one in the presence of cobalt ions, indicating that the tetramerization property conferred by the TetrHis motif should be transferable to a wide range of scFvs with various binding specificities. This suggests that the TetrHis motif could be used as a chemically switchable fiducial and size enhancer for cryo-EM structural determination of scFv complexes, providing a tetravalent platform with a fixed orientation of bound particles.

Although the TetrHis motif requires additional scFv framework residues to function, the crystal structure of scFv 2A2 provides information about the parts of the scFv framework that are required to stabilize the motif, and parts that are dispensable and can be engineered to improve, for example, the stability of the monomer to increase bacterial expression levels. In addition, the structural information can be used to design mutations that further improve the stability range of the cobalt-stabilized tetramer by creating additional interactions.

Metal−protein nanohybrid materials are a class of fast emerging functional nanomaterials with a broad range of potential applications such as biomineralization, catalysis, drug delivery, tumor imaging and therapy, and others[30]. In the era of protein design, the TetrHis motif could also be used as a template to design dense metal clusters embedded within larger protein structures, or metal-dependent self-assembling repeating units, thereby generating new types of hydrogels, or even protein-based metal-organic frameworks (MOF).

## Materials and methods

**Construct design, expression, and purification of scFv 2A2**. ScFv 2A2 was designed by inserting the CDRs sequences described for anti-ceramide antibody (https://patents.google.com/patent/US20190389970A1/en and[17]) into a mouse scFv scaffold with a $(Gly_4Ser)_4$ linker and by adding a sortase motif LPETG followed by a 6x-Histag and a Twin-Strep-tag[31] at the C-terminus. The corresponding synthetic gene was synthesized (Eurofins Genomics) and cloned into Drosophila melanogaster S2 expression vector for scFv[32]. This resulted in a mature secreted construct with the following primary structure: $V_H$ - $(Gly_4Ser)_4$ - $V_L$ – sortase – $His_6$ – TwinStrep. Drosophila S2 cells were transfected as reported previously[33], amplified, and scFv expression was induced with 4 µM $CdCl_2$ at a density of ~10x $10^6$ cells per ml for 6-8 days for large-scale production. The protein was purified from the supernatant by affinity chromatography using a Strep-Tactin resin (IBA) according to manufacturer's instructions followed by SEC on a Superdex200 column (GE Healthcare). Pure monomeric scFv was concentrated to 8.5 mg.ml$^{-1}$ and frozen at −80 °C.

**Construct design, expression, and purification of scFv 748**. ScFv 748 was designed by performing a structural alignment of the scFv 2A2 monomer onto the crystal structure of anti-FLAG M2 Fab[19] (PDB ID 2G60), and replacing the CDR loops from scFv 2A2. The synthetic gene corresponding to the designed sequence was ordered from Genecust, custom cloned into a pET26b vector with a pelB leader sequence for periplasmic expression in E. coli. The plasmid was transformed into BL21 DE3 E. coli cells and grown on kanamycin-supplemented plates. A single colony was used to inoculate liquid cultures overnight. The scFv was then expressed by overnight incubation under shaking at 18 °C following 0.25 mM IPTG induction of 4 L of LB after the OD600nm reached 0.6. Cells were harvested by centrifugation and the resulting cell pellets were resuspended in 20 mM Tris, pH 7.5, 150 mM NaCl. Cells were lyzed by sonication, and the lysate was centrifuged for 25 min at 4 °C and 50,000×g to remove cell debris. The supernatant was loaded onto a column containing 2 ml of pre-equilibrated Ni-NTA superflow (QIAGEN). After extensive washes, the protein was eluted in 20 mM Tris, pH 7.5, 150 mM NaCl, 400 mM imidazole. The eluate was then diluted to <200 mM imidazole concentration and loaded onto a column containing 2 ml of pre-equilibrated Strep-Tactin resin (IBA). After washing, the protein was eluted using 2.5 mM desthiobiotin. Concentrated eluate was then subjected to size exclusion chromatography on a S200 column equilibrated in 50 mM HEPES pH 7.5 and 150 mM NaCl.

**X-ray crystallography**. Crystallization was carried out by vapor diffusion using a Cartesian Technologies pipetting system[34]. scFv 2A2 was concentrated to 8.5 mg/ml in 50 mM HEPES pH 7.5 and 150 mM NaCl. The protein crystallized at 20 °C in mother liquor containing 0.01 M cobalt chloride, 0.1 M MES, pH6.5, 1.8 M ammonium sulfate after ~5–10 days. Crystals were frozen in liquid nitrogen after being cryoprotected with mineral oil. Diffraction data were collected at X06SA beamline of the Swiss Light Source (SLS), Villigen, Switzerland, with a wavelength of 0.999 Å. A single dataset at 2.5 Å resolution was obtained. All data were automatically processed by xia2[35]. Structural determination was initiated by molecular replacement using an homology model of scFv 2A2 obtained via SWISS-MODEL[36] as a search model in PHASER[37]. The solution was subjected to repetitive rounds of restrained refinement in PHENIX[38] and Autobuster[39] and manual building in

COOT[40]. TLS parameters were included in the final round of refinement. Data collection and refinement statistics are provided in Table 1, and the final refined coordinates and structure factors have been deposited in the PDB with accession code 8CGE.

**Dynamic light scattering**. DLS measurements were performed at 20 °C using the Malvern Zetasizer Nano S instrument (Malvern, Worcestershire, England) equipped with a Peltier temperature controller. Data analysis was performed using the Zetasizer Nano S DTS software package. The hydrodynamic radii calculated from the intensity size distributions were compared to theoretical values obtained from structural models of scFv 2A2 using the HullRad web server[41].

**Small-angle x-ray scattering**. Small-angle X-ray scattering measurements of *scFv 2A2* at 2.1, 4.2, and 8.5 mg/ml were performed at the SWING beamline of the French national synchrotron facility (SOLEIL). Data was collected at 15 °C, a wavelength of 1.0332 Å and a sample-to-detector distance of 1.99 m. For batch measurements, the scattering from the buffer alone was measured before and after each sample measurement and was used for background subtraction with PRIMUS from the ATSAS package[42]. For SEC-SAXS measurements, scFv 2A2 samples at 8.5 mg/ml were loaded onto a Superdex200 5/150 column (GE Healthcare) previously equilibrated in 50 mM HEPES pH 7.5 and 150 mM NaCl with or without addition of 5 mM $CoCl_2$. The measured SAXS images were normalized to the transmitted intensity and azimuthally averaged by using the in-house software Foxtrot (https://www.synchrotron-soleil.fr/en/beamlines/swing). The resulting SAXS curves were analyzed using CHROMIXS[43] and PRIMUS[44].

Additional SAXS measurements were performed on beamline BM29 at the European Synchrotron Radiation Facility (ESRF), Grenoble, France. Samples were kept at 20 °C and data were collected at a wavelength of 0.0995 nm and a sample-to-detector distance of 1 m. 1D scattering profiles were generated and buffer subtraction was carried out by the automated data processing pipeline available at BM29.

**Molecular dynamics and ensemble optimization of scFv 2A2**. Classical explicit solvent molecular dynamics simulations were used to generate conformational ensembles of scFv 2A2 in its monomeric and cobalt-stabilized tetrameric states, in order to fit the SAXS data. The starting models were extracted from the scFv 2A2 crystal structure and missing (disordered) residues from the (Gly$_4$Ser)$_4$ were manually added in Coot. Both systems were simulated in GROMACS[45] using the amber99SBws forcefield[46], which was designed to reproduce the properties of intrinsically disordered proteins. In order to maintain the geometry of the 8 cobalt binding sites in the tetrameric form, the metal centers were replaced by zinc and harmonic distance restraints (cutoff 2.5 Å) were applied between each Zn2+ ion and the coordinating residues. The zinc (II) parameters are available as part of the standard amber99SBws forcefield distribution. At the beginning of each simulation, the protein was immersed in a box of TIP4P2005 water, with a minimum distance of 1.0 nm between protein atoms and the edges of the box. The genion tool was used to add 150 mM NaCl. Long-range electrostatics were treated with the particle-mesh Ewald summation. Bond lengths were constrained using the P-LINCS algorithm. The integration time step was 5 fs. The v-rescale thermostat and the Parrinello–Rahman barostat were used to maintain a temperature of 300 K and a pressure of 1 atm. Each system was energy minimized using 1,000 steps of steepest descent and equilibrated for 500 ps with restrained protein heavy atoms prior to production simulations. Multiple independent MD trajectories were calculated for each system, for a total aggregated simulation time of ≈2.8 µs for the monomer and ≈ 2.0 µs for the tetramer. Snapshots were extracted every 1 ns from each trajectory, leading to the generation of 2760 models of the monomer and 2042 models of the tetramer. For each model from both ensembles, theoretical SAXS patterns were calculated with CRYSOL[47] and ensemble optimization fitting was performed with GAJOE[29].

**Reporting summary**. Further information on research design is available in the Nature Portfolio Reporting Summary linked to this article.

## Data availability
The final refined coordinates and structure factors of scFv 2A2 crystal structure (Supplementary Data 1) have been deposited in the PDB with accession code 8CGE.

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

## Acknowledgements

We acknowledge the ESRF, PSI, and SOLEIL for provision of synchrotron radiation facilities and we would like to thank the staff of beamline X06SA at the PSI for assistance with crystal testing and data collection, the staff of BM29 beamline at ESRF and the staff of SWING beamline at SOLEIL for assistance with SAXS data acquisition.

## Author contributions

R.D.H. designed, expressed, purified and crystallized scFv 2A2 preparations with the help of L.C. and F.H.; F.H. collected x-ray crystallography data. C.L. solved and refined the x-ray structure. C.L. and A.M. measured SAXS data. C.L. analyzed SAXS data and performed the computational studies. C.L. and P.C. prepared anti-Flag M2 scFv with the help of A.F. C.L. and A.F. measured and analyzed DLS data. C.L. and S.G. wrote the manuscript and jointly supervised the overall project.

## Competing interests

The authors declare no competing interests.
