## [Peer Review File · Communications Chemistry]

Reviewers' comments:

Reviewer #1 (Remarks to the Author):

The manuscript of Healey et al. provides first crystallographic structure and dynamics of a cobalt-dependent 6XHis tetramerization motif (TetrHis). The claims are certainly novel and potentially impactful. All these results seem to deserve publishing in Communications Chemistry. However, the manuscript is still in premature stage. Authors should add and modify following points to improve the report.

Major Issues:

Consistent with the crystallographic data, the authors claim that scFv 2A2 exists as a cobalt-stabilized tetramer in solution. However, the data do not doubtlessly demonstrate that the formation of a tetramer in solution is mediated by TetrHis motif. I can only urge the authors to do assays using a mutant lacking the TetrHis motif as a negative control.

Minor Issues:

(1) Lines 248~251: SEC-SAXS data indicate that scFv 2A2 forms only monomers or dimers in solution, which is inconsistent with crystallographic and MD simulation data. Authors should discuss about this.

(2) All figures included in the main text and supplementary materials are very blurry, low-resolution images. It is strongly suggested to improve the resolution by referring to the author guidelines provided by the journal. Also, the font size in some figures is too small to read.

(3) All figures must be cited in the text and must be quoted in order. However, there are cases where citations are omitted from the text or the order is jumbled.

(4) Label the histidine residues coordinating the nickel ions in Figure 3E-I.

(5) Please include Ramachandran plot statistics in Table 1. Also indicate the number of reflections for the working and test data sets, respectively.

(6) Indicate the crystallographic 2-fold axis in Figure 1B.

(7) In addition to cobalt ions, 15 hydrogen bonds found at the VL-VL interface play crucial roles in stabilization of scFv 2A2 tetrameric form. Thus, a close-up view of this extensive hydrogen bond network has to be added to Figure 1.

Reviewer #2 (Remarks to the Author):

The manuscript by Cedric Leyrat et al. entitled " Structure and dynamics of TetrHis, a novel metal-dependent polyhistidine tetramerization motif " is devoted to the important biochemical problem of the

structure and dynamics of a metal-dependent tetramerization motif with minimal sequence ETGHHHHHWSHPQ (TetrHis) observed in the cobalt-bound crystallographic structure of a single chain variable fragment (scFv) 2A2 antibody. This TetrHis contains a common polyhistidine (6XHis) fusion tag used for protein purification.

The topic of this study is within the scope of the Communications Chemistry since structure and dynamics in complex proteins are studied using crystallography, solution SAXS and molecular dynamics simulation atomic details and some perspectives are provided. The major innovation is the clarification and enumeration of the TetrHis motif metal-dependent tetramerization property and metal coordination sites to stabilize the scFv complex formation.

Usually, the 6XHis motif behaves as a flexible terminal loop, adopting various conformations, and is thus extremely rare resolved in crystallography structure. The authors made a protein data bank (PDB) structure motif search for 4His in 250 structures, and containing any metal ions returns 81 entries.

The paper reads perfectly as it is written with a clear explanation of the procedure and the context of results that enlighten the TetrHis motif metal-dependent tetramerization property in both crystal and solution, by SAXS measurements and MD simulations. The tetrameric assembly possesses a unique metal dependent β -stranded architecture with Co^{2+} ions binding sites clustered within a small region of space, which is further stabilized by the formation of VL-11 β -sheet dimers at the VL-VL interfaces, a common type of weak homotypic antibody interface. The authors further propose this TetrHis motif tetramerization property could be transferred to other scFvs provided, thus enable the TetrHis tag use as a chemically switchable fiducial and size enhancer for cryo-EM structural determination of scFv complexes, and others.

In general, I like the idea of this manuscript, the results presented in the manuscript are interesting and timely, and I have only several minor suggestions for further improvements.

Comments:

Results:

Q1. Page 3-8, In this reviewer opinion, the PDB structure examination and SAXS measurements for different buffer conditions are not enough when you are considering the protein concentrations (Table 2). It seems lack of a rationale on how to better stabilize the TetrHis tetramer in scFv 2A2, to make better control of this finding, a deeper understanding of how TetrHis tetramers are stabilized, especially under different environmental conditions, is required.

Q2. Page 6-8, The SAXS analysis of the study focus on the Guinier region fitting R_g values for the corresponding particle size, the information is too limited to interpret and the Kratky plots (Fig 4B) show that the protein complex folding may not well folded in the present solution condition. The research on the mechanism and influencing factors of TetrHis tetramer formation is still insufficient, and the various factors that affect the formation of TetrHis tetramer, such as metal ion concentration, pH value, reaction time, etc., need to be further explored.

Q4. Page 9, The author ran the scFv complex MD simulations in which the cobalt ions replaced by zinc ions, and/or restraint used. This may cause the discrepancy for the interaction of the TetrHis sequence with cobalt sites, which may affect the stability and tetramerization properties and the open/close state population.

Discussion:

Q5. Page 10,: Since the study used a high concentration of Co²⁺ ions to induce the formation of tetramers, “successful crystallization in the presence of a high concentration of cobalt ions”, further examination of the metal concentration dependence and studies on the tetramerization properties of TetrHis sequences at a rationale low to high target concentrations are required.

Q6. Page 16, the Trp 277 residue twice labeled in Figure 1E and Figure presentation is difficult to evaluate.

In general, the discovery and application potential of this study is significant valuable, but there are some limitations, and further research is encouraged to explore its more application value.

Reviewer #3 (Remarks to the Author):

This paper describes the structural, computational and biophysical characterization of a single chain antibody (scFv) harboring a long tail that includes a 6XHis tag. The authors demonstrate that this system tetramerizes in the presence of cobalt(II) through the assembly of a structural motif formed by the 6xHis stretches from the four chains, stabilized by 8 cobalt(II) ions. The tetrameric form of the tagged scFv exists both in the crystal and in solution. The authors further show that the present structural arrangement is unprecedented in the PDB.

The overall experimental approach is quite solid and provides convincing evidence that tetramerization is linked to the selection of specific configurations of the system, which has relevant inter-domain dynamics, by the cobalt(II) ions.

A main contention of the authors is that other scFv's can tagged similarly to the present one, leading to a novel methodology for the design of self/assembling metal-organic frameworks. However, this is not adequately supported by the current data. In fact, given the widespread use of the His-tag in protein purification, the fact that this the first such observation suggests that obtaining a functional multimerization platform may be far from obvious. Even if such multimeric states would not crystallize, detection of multimers would have occurred with routine essays such as LS or in size-exclusion chromatography. In short, I believe that some experimental evidence corroborating the transferability of the size-enhancing properties of the tag is needed to warrant publication.

Minor points:

- The fonts in the Figures are often too small (e.g., Figs 3 and 5)
- There needs to be a description of what the authors mean by "linear motif"
- The parameters for the zinc(II) ion in MD simulations are not specified

Reviewer #4 (Remarks to the Author):

In this paper, Healey et al reported a Co²⁺ binding-protein tetramer and its dynamic open/close structural transformation. The results are interesting and should be published; however, paper organization and data presentation is basic and short of sophistication of a research article, making it read more like an experimental report rather than research article. The major revisions should be made before it can be further considered for publication in Communication Chemistry and suggestions are listed as follows:

1. in introduction part, I am not very clear the purpose of paragraph 2 that show some research results in PDB database. The author should provide a concluding statement summarizing previous studies in this field and highlighting core limitations or problems in that this paper aims to address. This will help readers understand the novelty of this paper.
2. the paragraph 4 in introduction is also meaningless, which should be shorted and then appended to paragraph 3.
3. the single crystal structure of cobalt-bound scFv 2A2 tetramer is impressive, what is colour of obtained crystal particles containing Co²⁺, is it different to the color of free Co²⁺ ions in solution? The crystals consisted of high content of β -sheet, is crystals stable in the air? Can the crystals be used as a material for electrocatalysis applications?
4. the author states that 'the structure of this novel metal-dependent tetramerization motif might provide a useful starting point for designing metal-loaded biomaterials, with potential applications in the fields of biosensors, bioanalytical devices, or biocatalyst', those mentioned applications are based on the electrochemical property of crystals, therefore, redox C-V curve of Co containing protein crystals should be provided.
5. Research results in PDB database were shallow results and could not strongly support the novelty and significant of the sequence motif, which is determined by the intrinsic feature of physical and chemical structures and functions. Therefore, reviewer suggested that discussions on this part should be shorted or even removed, and how metal-protein ligands interaction contributed to the complex architectures should be more emphasized.
6. in figure 4, the presentation of SAXS data confused readers. For example, in figure 4A, each sample has three different SAXS curves, each curve should be marked. Correspondingly, 'the shape of the measured SAXS profiles changed slightly, with the appearance of a small bump at intermediate Q, particularly visible at high protein concentration, suggesting significant hollowness of the scattering object. This feature was consistent with the presence of a large void within the tetrameric scFv structure (Figure 1B).', this description is also very confused, what is value of the intermediate Q, and what is the size of the void within the tetrameric?
7. in figure 5B, the simulations maybe not relaxed enough for partially dissociated tetramer (black line), this could be not a good positive evident to support that close conformation is necessary to formation of tetramer. This figure could be removed to the SI?

8. Other alternative technologies as such circular dichroism spectrum (CD) and dynamic light scattering (DLS) should be considered for measurements to support the formation process of tetramer in solution.

Response to the reviewers' comments:

We would like to thank the reviewers for providing constructive feedback on our manuscript. We have now performed additional SAXS measurements and ensemble analysis on scFv 2A2 that strengthen our initial claims regarding the formation of tetramers in solution. We additionally performed a cobalt titration using dynamic light scattering, in which we measured scFv 2A2 dimensions in solution, and also incorporated a negative control scFv that doesn't contain the His-tag (as requested by reviewer #1). Finally, before submitting the manuscript we had ordered synthetic genes of 2 scFvs based on the TetrHis motif and asked for them to be cloned into a pET26 for periplasmic expression in E. coli. We expressed and purified 1 of them (scFv 748), characterized the influence of 5 mM of Co²⁺ on its measured SAXS profile, and were able to confirm that it behaves similarly to scFv 2A2. We have extensively rewritten various parts of the manuscript and hope you will find that all the points raised have been addressed. Please find below a detailed response to each comment.

Reviewer #1 (Remarks to the Author):

The manuscript of Healey et al. provides first crystallographic structure and dynamics of a cobalt-dependent 6XHis tetramerization motif (TetrHis). The claims are certainly novel and potentially impactful. All these results seem to deserve publishing in Communications Chemistry. However, the manuscript is still in premature stage. Authors should add and modify following points to improve the report.

Major Issues:

Consistent with the crystallographic data, the authors claim that scFv 2A2 exists as a cobalt-stabilized tetramer in solution. However, the data do not doubtlessly demonstrate that the formation of a tetramer in solution is mediated by TetrHis motif. I can only urge the authors to do assays using a mutant lacking the TetrHis motif as a negative control.

We have now performed a dynamic light scattering experiment in which we measured the effect of increasing concentrations of cobalt ions on the intensity size distributions of scFv 2A2. We included a negative control that has a C-terminal twin strep tag but no sortase motif or 6XHis. These data show that the TetrHis motif is responsible for inducing changes in the hydrodynamic radius that are consistent with a monomeric to tetrameric transition. Unexpectedly, the negative control showed an opposite trend, seemingly transitioning from a larger species down to a monomer in the presence of cobalt. Although we didn't comment on this result in the paper as we feel this is outside the scope of the study, this could mean that the negative control scFv exists as an open state conformer and/or in an oligomeric form in solution, and that cobalt is able to bind the scFv in a way that stabilizes the close state monomer, possibly through the CDRs. Whatever the reason, the observed behavior is very different from scFv 2A2, and the control scFv has hydrodynamic radius consistent with monomer at all cobalt concentrations tested.

Minor Issues:

(1) Lines 248~251: SEC-SAXS data indicate that scFv 2A2 forms only monomers or dimers in solution, which is inconsistent with crystallographic and MD simulation data. Authors should discuss about this.

You are correct that the molecular weight of 30-60kDa estimated from the SEC-SAXS data in the presence of cobalt might suggest that the protein only form monomers and dimers. However, our analyses show that this is in fact due to the dynamic equilibrium between monomer and tetramer resulting in the presence of mixed populations in the SEC peak. This is clearly demonstrated by the EOM ensemble analysis of the SEC-SAXS data, but also now by the EOM analysis of the solution-based SAXS data in the presence of cobalt and nickel ions, which we hadn't done in the previous version of paper. In addition, we have added a figure panel with intramolecular distance distributions calculated from the SAXS data, which showed 2 overlapping peaks at 1 and 2 mg/ml of protein concentration in the presence of cobalt, also indicating an equilibrium between 2 different species.

(2) All figures included in the main text and supplementary materials are very blurry, low-resolution images. It is strongly suggested to improve the resolution by referring to the author guidelines provided by the journal. Also, the font size in some figures is too small to read.

We have now uploaded high-resolution versions of the images separately, which should solve the problem. We also increased the font size in various figure panels.

(3) All figures must be cited in the text and must be quoted in order. However, there are cases where citations are omitted from the text or the order is jumbled.

This should now be fixed.

(4) Label the histidine residues coordinating the nickel ions in Figure 3E-I.

done

(5) Please include Ramachandran plot statistics in Table 1. Also indicate the number of reflections for the working and test data sets, respectively.

done

(6) Indicate the crystallographic 2-fold axis in Figure 1B.

done

(7) In addition to cobalt ions, 15 hydrogen bonds found at the VL-VL interface play crucial roles in stabilization of scFv 2A2 tetrameric form. Thus, a close-up view of this extensive hydrogen bond network has to be added to Figure 1.

done

Reviewer #2 (Remarks to the Author):

The manuscript by Cedric Leyrat et al. entitled " Structure and dynamics of TetrHis, a novel metal-dependent polyhistidine tetramerization motif " is devoted to the important biochemical problem of

the structure and dynamics of a metal-dependent tetramerization motif with minimal sequence ETGHHHHHWSHPQ (TetrHis) observed in the cobalt-bound crystallographic structure of a single chain variable fragment (scFv) 2A2 antibody. This TetrHis contains a common polyhistidine (6XHis) fusion tag used for protein purification.

The topic of this study is within the scope of the Communications Chemistry since structure and dynamics in complex proteins are studied using crystallography, solution SXAS and molecular dynamics simulation atomic details and some perspectives are provided. The major innovation is the clarification and enumeration of the TetrHis motif metal-dependent tetramerization property and metal coordination sites to stabilize the scFv complex formation.

Usually, the 6XHis motif behaves as a flexible terminal loop, adopting various conformations, and is thus extremely rare resolved in crystallography structure. The authors made a protein data bank (PDB) structure motif search for 4His in 250 structures, and containing any metal ions returns 81 entries.

The paper reads perfectly as it is written with a clear explanation of the procedure and the context of results that enlighten the TetrHis motif metal-dependent tetramerization property in both crystal and solution, by SAXS measurements and MD simulations. The tetrameric assembly possesses a unique metal dependent β -stranded architecture with Co^{2+} ions binding sites clustered within a small region of space, which is further stabilized by the formation of VL-11 β -sheet dimers at the VL-VL interfaces, a common type of weak homotypic antibody interface. The authors further propose this TetrHis motif tetramerization property could be transferred to other scFvs provided, thus enable the TetrHis tag use as a chemically switchable fiducial and size enhancer for cryo-EM structural determination of scFv complexes, and others.

In general, I like the idea of this manuscript, the results presented in the manuscript are interesting and timely, and I have only several minor suggestions for further improvements.

Comments:

Results:

Q1. Page 3-8, In this reviewer opinion, the PDB structure examination and SAXS measurements for different buffer conditions are not enough when you are considering the protein concentrations (Table 2). It seems lack of a rationale on how to better stabilize the TetrHis tetramer in scFv 2A2, to make better control of this finding, a deeper understanding of how TetrHis tetramers are stabilized, especially under different environmental conditions, is required.

We have now performed additional DLS and SAXS experiments, and systematically fitted our monomer-tetramer ensemble against the SAXS profiles, providing new insights into the cobalt and protein concentration dependence of tetramerization. We also showed that nickel (II) ions induce similar proportions of tetramers in solution.

Q2. Page 6-8, The SAXS analysis of the study focus on the Guinier region fitting R_g values for the corresponding particle size, the information is too limited to interpret and the Kratky plots (Fig 4B) show that the protein complex folding may not well folded in the present solution condition.

We agree that the initial SAXS data analysis focused on R_g values was too limited. For this reason we have now added a comparison of $p(r)$ distance distribution functions of the protein in the absence or in the presence of cobalt ions (now figure 5C). Regarding the Kratky plots, they do contain valuable information regarding the type of protein structure and dynamics present in the data, and have the added advantage of not relying on 3D models of proteins or other assumptions. In this case the Kratky plots are consistent with a transition from a mostly globular protein to something that is multidomain (for example because of a change in quaternary structure). The interpretation is based on the analysis of different types of proteins using Kratky plots (see figure 2 in Curr Protein Pept Sci. 2012 Feb; 13(1): 55–75. doi: 10.2174/138920312799277901 or image from Stanford Synchrotron Radiation Lightsource https://www-ssrl.slac.stanford.edu/smb-saxs/sites/default/files/images/kratky_plot_example_2019_600.jpg).

The research on the mechanism and influencing factors of TetrHis tetramer formation is still insufficient, and the various factors that affect the formation of TetrHis tetramer, such as metal ion concentration, pH value, reaction time, etc., need to be further explored.

As stated above, the new DLS, SAXS measurements and analyses now provide insights into the metal ion and protein concentration dependence of tetramerization. We have not tested pH dependence, but it is well-known that the optimum pH for binding of histidine residues to metals is between 7 and 8. Regarding reaction time, the SEC-SAXS profile shows a widening peak which suggests a fast exchange between monomers and tetramers. Based on the DLS measurements, we know that the observed changes are faster than the delay between measurements (a few minutes).

Q4. Page 9, The author ran the scFv complex MD simulations in which the cobalt ions replaced by zinc ions, and/or restraint used. This may cause the discrepancy for the interaction of the TetrHis sequence with cobalt sites, which may affect the stability and tetramerization properties and the open/close state population.

You are correct that replacing the cobalt ions by zinc in the simulation can impact the stability and tetramerization properties of the scFv. Indeed, we observed that in the absence of distance restraints, zinc ions progressively unbind and the tetramer starts to dissociate, especially on longer timescales (> 500 ns). For this reason, we implemented distance restraints between each metal ion and its coordinating residues, which manages to keep the tetramer close to the crystallographically observed conformation (but effectively sample loop and disordered tail conformations, which is the main goal here).

In this study the only purpose of the MD simulations is to obtain ensembles of PDB files that capture (as much as possible) the physically accessible conformations of the scFv in solution. We don't actually use the MD simulations of the tetramer to draw any sort of conclusions about the scFv, but instead pool all the observed conformations in an ensemble that is then used to analyze the SAXS data. In this way, we let the genetic algorithm implemented in EOM decides which conformations are actually experimentally relevant based on the SAXS data, and we monitor the quality of the fit. Consequently, the MD simulations have no direct effect on which proportions of open/close state monomers, or tetramers, are selected in the optimized ensembles fitting the SAXS data.

Discussion:

Q5. Page 10,: Since the study used a high concentration of Co 2+ ions to induce the formation of

tetramers, “successful crystallization in the presence of a high concentration of cobalt ions”, further examination of the metal concentration dependence and studies on the tetramerization properties of TetrHis sequences at a rationale low to high target concentrations are required.

You are correct that this aspect was not sufficiently well-characterized in the previous version of our paper. As stated above, the additional DLS and SAXS measurements, as well as the new analyses ($p(r)$ functions and new EOM fits), now provide insights into the cobalt and protein concentration dependence of tetramerization. We have also updated the discussion to reflect the new results.

Q6. Page 16, the Trp 277 residue twice labeled in Figure 1E and Figure presentation is difficult to evaluate.

Indeed, Trp277 is twice labeled in figure 1E. This is because two Trp277 residues (coming from the strep tag of 2 different chains) interact together and with residues from the scFv framework (e.g. Phe241). We have now modified the figure panel and changed the color of the framework residues in an attempt to make the figure clearer, and highlight which residues come from the TetrHis motif versus the rest of the scFv.

In general, the discovery and application potential of this study is significant valuable, but there are some limitations, and further research is encouraged to explore its more application value.

Thank you for these positive comments, we hope you will find that the new data and analyses now provide a more comprehensive view of the cobalt dependent tetramerization mechanism, and of how it can be transferred to other scFv molecules.

Reviewer #3 (Remarks to the Author):

This paper describes the structural, computational and biophysical characterization of a single chain antibody (scFv) harboring a long tail that includes a 6XHis tag. The authors demonstrate that this system tetramerizes in the presence of cobalt(II) through the assembly of a structural motif formed by the 6xHis stretches from the four chains, stabilized by 8 cobalt(II) ions. The tetrameric form of the tagged scFv exists both in the crystal and in solution. The authors further show that the present structural arrangement is unprecedented in the PDB.

The overall experimental approach is quite solid and provides convincing evidence that tetramerization is linked to the selection of specific configurations of the system, which has relevant inter-domain dynamics, by the cobalt(II) ions.

A main contention of the authors is that other scFv's can tagged similarly to the present one, leading to a novel methodology for the design of self/assembling metal-organic frameworks. However, this is not adequately supported by the current data. In fact, given the widespread use of the His-tag in protein purification, the fact that this the first such observation suggests that obtaining a functional multimerization platform may be far from obvious. Even if such multimeric states would not crystallize, detection of multimers would have occurred with routine essays such as LS or in size-exclusion chromatography. In short, I believe that some experimental evidence corroborating the transferability of the size-enhancing properties of the tag is needed to warrant publication.

You are correct that the 1st version of our paper didn't provide any evidence in support of the transferability of the tetramerization property conferred by the TetrHis motif. We have now characterized using SAXS a new construct that incorporates the same framework and C-terminal tags as the original scFv, but with different CDR loops. As expected from the crystallographic data (given the location of the CDRs on the opposite face of the scFv), the new scFv had very similar SAXS profiles to the original one, both in the presence and absence of cobalt ions, and a similar proportion of tetramers was induced by cobalt based on the ensemble optimization analysis.

Regarding the fact that this is the 1st observation of such metal-dependent tetramerization, we think this is not so surprising because it would have required a combination of fairly low probability events: (1) it is uncommon to put purified proteins in buffer containing free cobalt or nickel ions. In biochemistry labs, these ions are mostly used to regenerate NiNTA, TALON beads etc... (2) a C-terminal sortase motif followed by a 6XHis and Twin strep tag is also quite uncommon in protein constructs. (3) the tags have to be put on a scFv in which the framework sequence is compatible with packing of the motif in the metal bound state. This requires a few framework residues such as Pro238, Phe241, Met243, Lys261 and Glu263 to be conserved or replaced by residues that can support similar interactions. For example, in the control anti-ADIPOR scFv used in the DLS experiment, Met243 is replaced by a serine, which may lower the stability of the tetrameric state.

These combined requirements provide a reasonable explanation as to why the TetrHis motif was not seen before.

Minor points:

- The fonts in the Figures are often too small (e.g., Figs 3 and 5)

This should now be better, also we have uploaded high resolution figures separately so things should be less blurry.

- There needs to be a description of what the authors mean by "linear motif"

We added this short description at lines 192-194: We found only five occurrences of linear **metal-binding** motifs (i.e. short amino acid sequences typically ranging from 3 to 10 residues in length) out of the 27 structures, all of which involve one or two bound nickel ions, stabilizing either a dimeric or trimeric assembly (figure 3E-I).

- The parameters for the zinc(II) ion in MD simulations are not specified

This sentence was added to the MD simulation part of the methods section: "The zinc (II) parameters are available as part of the standard amber99SBws forcefield distribution."

Reviewer #4 (Remarks to the Author):

In this paper, Healey et al reported a Co²⁺ binding-protein tetramer and its dynamic open/close structural transformation. The results are interesting and should be published; however, paper organization and data presentation is basic and short of sophistication of a research article, making it read more like an experimental report rather than research article. The major revisions should be made before it can be further considered for publication in Communication Chemistry and

suggestions are listed as follows:

1. in introduction part, I am not very clear the purpose of paragraph 2 that show some research results in PDB database. The author should provide a concluding statement summarizing previous studies in this field and highlighting core limitations or problems in that this paper aims to address. This will help readers understand the novelty of this paper.

The purpose of paragraph 2 is to state that despite the widespread use of poly histidine tags in protein structural biology, these sequences don't usually adopt a stable, well-defined three-dimensional structure. We would have liked to cite previous studies regarding the multimerization of poly histidine sequences, but we couldn't find any. We tried to modify paragraph 2 and 3 to help readers understand the novelty of the paper better, as suggested. We also added this sentence: "However, the identification and characterization of a metal-dependent tetramerization motif utilizing a polyhistidine sequence in scFvs represents a novel and unexplored area of research."

2. the paragraph 4 in introduction is also meaningless, which should be shorted and then appended to paragraph 3.

We have now merged paragraph 3 and 4.

3. the single crystal structure of cobalt-bound scFv 2A2 tetramer is impressive, what is colour of obtained crystal particles containing Co^{2+} , is it different to the color of free Co^{2+} ions in solution? The crystals consisted of high content of β -sheet, is crystals stable in the air? Can the crystals be used as a material for electrocatalysis applications?

The crystals of the scFv 2A2 tetramer were harvested from a crystal screen plate and soaked in mineral oil before freezing in liquid nitrogen. Only a single crystal provided good quality diffraction, which we used to solve the structure. We don't routinely keep pictures of protein crystals, but in this case we don't recall that these crystals had any special colour features. The number of cobalt ions in the crystal is very small compared to the amount of protein. Protein crystals are generally not stable in air as they are tiny, contain a lot of water and dehydrate very quickly. We don't believe that they could be used as a material for electrocatalysis applications.

4. the author states that 'the structure of this novel metal-dependent tetramerization motif might provide a useful starting point for designing metal-loaded biomaterials, with potential applications in the fields of biosensors, bioanalytical devices, or biocatalyst', those mentioned applications are based on the electrochemical property of crystals, therefore, redox C-V curve of Co containing protein crystals should be provided.

You are correct that these applications rely on the electrochemical properties of crystals. We wrote that the structure of the tetramerization motif might provide a starting point for designing metal-loaded biomaterials, and we mean that it could be a starting point to design different materials, but only a starting point. We don't expect potential applications in the fields of biosensors, bioanalytical devices, or biocatalyst to come out of scFv 2A2 crystals, but we were thinking about crystals that would incorporate only the metal-binding motif (e.g. short, potentially chemically modified peptides or derivatives). To be honest, the reason we mentioned such potential applications is to provide keywords that may increase the likelihood of the paper being read by chemists interested in developing such materials. We have now removed the mention of these potential applications from the abstract in order to fit within the word limit, but left the statement at the end of the discussion.

5. Research results in PDB database were shallow results and could not strongly support the novelty and significance of the sequence motif, which is determined by the intrinsic features of physical and chemical structures and functions. Therefore, reviewer suggested that discussions on this part should be shortened or even removed, and how metal-protein ligands interaction contributed to the complex architectures should be more emphasized.

In the protein structural biology field, it is quite common when reporting a new structure to determine if similar structures exist in the PDB and make comparisons. In this case, the metal bound tetrHis structure was very unusual and we felt it was important to show what the most similar structures present in the PDB look like.

6. In figure 4, the presentation of SAXS data confused readers. For example, in figure 4A, each sample has three different SAXS curves, each curve should be marked.

We have now labeled each curve with the protein concentration, and also added 4 new SAXS curves measured at lower protein concentrations with and without cobalt ions in the buffer.

Correspondingly, 'the shape of the measured SAXS profiles changed slightly, with the appearance of a small bump at intermediate Q, particularly visible at high protein concentration, suggesting significant hollowness of the scattering object. This feature was consistent with the presence of a large void within the tetrameric scFv structure (Figure 1B).', this description is also very confused, what is the value of the intermediate Q, and what is the size of the void within the tetrameric?

In biological SAXS, the intermediate Q range generally refers to Q comprised between 0.1 and 0.3 Å⁻¹ where you get most information about the shape of the macromolecule.

We have tried to make this description clearer for the reader:

"In the presence of Co²⁺ or Ni²⁺ ions, the measured SAXS profiles showed a noticeable change of slope between Q ≈ 0.1-0.2 Å⁻¹, particularly visible at high protein concentration (Figure 5A), suggesting significant hollowness of the scattering object. This feature was consistent with the presence of a large void within the tetrameric scFv structure (Figure 1B), with the presence of a channel of 10-20 Å width in between the TetrHis motif and scFv domains."

7. In figure 5B, the simulations maybe not relaxed enough for partially dissociated tetramer (black line), this could be not a good positive evidence to support that close conformation is necessary to formation of tetramer. This figure could be removed to the SI?

The purpose of the MD simulations is to obtain ensembles of PDB files that capture (as much as possible) the physically accessible conformations of the scFv in solution. All conformations are then pooled together in an ensemble that is used to analyze the SAXS data. In this way, we let the genetic algorithm implemented in EOM decide which conformations are actually relevant based on the measured SAXS profiles. Consequently, the MD simulations have no direct effect on which proportions of open/close state monomers, or tetramers, are selected in the optimized ensembles fitting the SAXS data.

That being said, we don't mind removing this figure to the SI if necessary.

8. Other alternative technologies as such circular dichroism spectrum (CD) and dynamic light scattering (DLS) should be considered for measurements to support the formation process of tetramer in solution.

We considered both techniques to carry out additional experimental characterizations, but we thought that CD would not be very sensitive to tetramerization because there are not so many changes in secondary structure when going from monomer to tetramer. We have performed new DLS (and SAXS) measurements and analyses that provide insights into the metal ion and protein concentration dependence of tetramerization.

REVIEWERS' COMMENTS:

Reviewer #1 (Remarks to the Author):

All points raised have been adequately addressed by the authors. I recommend acceptance of the revised version of the manuscript.

Reviewer #2 (Remarks to the Author):

The authors of this paper have made revisions/changes accordingly to present clearly insights with evidence, on getting published impact of Communications Chemistry on the capability of a significant contribution to the field of question answering systems.

Reviewer #3 (Remarks to the Author):

The present revised version of the manuscript has improved greatly. My concerns about the transferability of the polyhistidine tetramerization motif have been adequately addressed. In my opinion, this contribution can be published as is.

Reviewer #4 (Remarks to the Author):

I recommend the publication in the present form, as the authors have responded my comments.